# Effects of psychogenic stress on some peripheral and central inflammatory markers in rats with the different level of excitability of the nervous system

I. G. Shalaginova [1] *, O. P. Tuchina[1], M. V. Sidorova[1], A. S. Levina[2], D. A-A. Khlebaeva[2], A. I. Vaido[2], N. A. Dyuzhikova[2]

1 Immanuel Kant Baltic Federal University, Kaliningrad, Russia, 2 Pavlov Institute of Physiology, Russian Academy of Sciences, Saint-Petersburg, Russia

* shalaginova_i@mail.ru

**Data Availability Statement:** All relevant data are within the manuscript and its Supporting Information files.

## Abstract

Patients with post-stress pathologies display the signs of inflammation in the peripheral blood as well as in the brain. The mechanisms of such post-stress neuroimmune changes, their contribution to the behavior, the relationship of the intensity of inflammation with genetically determined features have not been clarified. The goal of this work was to evaluate the dynamics of post-stress inflammation in the blood and hippocampus of rats which differ in level of excitability of the nervous system. Rats of two strains (high/low excitability threshold) were subjected to stress according to the K. Hecht protocol and their behavior, neutrophil: lymphocyte ratio and the number of Iba+ cells in the hippocampus were analysed 24 hours, 7 and 24 days after stress exposure. Highly excitable animals show an increase in anxiety-like behavior, in the number of neutrophils compared to lymphocytes as well as in the number of Iba1+ cells in CA1, CA3 and DG areas of the hippocampus in response to stress. Thus, hereditary high excitability of the nervous system is a possible risk factor for the development of post-stress pathologies.

## Introduction

It is well known that long-term stress is a risk factor for development of different psychopathologies such as anxiety and affective disorders [1,2]. Pathogenesis of these conditions is not well understood. The main hypothesis that explains the connection between stress and mental disorders suggests that disturbances in hypothalamic-pituitary axis and epigenetic alterations lead to neuronal and glial dysfunction [3]. There is strong evidence that patients with post-stress pathologies as well as animals which were exposed to acute and chronic stress display upregulation in cytokine signalling and other pro-inflammatory molecules in the peripheral blood and in the brain [4]. For instance, the levels of proinflammatory cytokines such as interleukin-1β (IL-1β), interleukin-6 (IL-6) and tumor necrosis factor alfa (TNFα) are significantly increased in the blood of patients with posttraumatic stress disorder (PTSD) compared with

**Funding:** This research was supported from the Russian Academic Excellence Project at the Immanuel Kant Baltic Federal University.

**Competing interests:** No authors have competing interests.

both the control and patients with a history of injury without the development of PTSD symptoms [5]. Prolonged action of stressors leads to the release of glucocorticoids in the blood and, as a result, to suppression of certain aspects of immune response such as shifting in neutrophil: lymphocyte (N:L) ratio [6,7].

In mice and rats an increase in the number of neutrophils and in some cases a decrease in the number of lymphocytes was observed 1–2 hours after stress exposure, as well as after the injection of corticosterone [6,7]. In mice, on day 8 after exposure to stress an increase in the level of monocytes circulating in the blood and increased anxiety-like behavior were observed. On day 24 mice showed abnormal behavior and increased mRNAs levels of proinflammatory cytokines (IL-1β, TNFα, and IL-6) in microglial cells, although there were no signs of inflammation in the blood [8]. There is also evidence of increased IL-1β, TNFα and IL-6 expression in the rat hippocampus in the PTSD model, and this effect persisted for two weeks after exposure to stress [9 for review].

The main source of these pro-inflammatory mediators in the brain is activated microglia [10]. Microglial cells originate from myeloid progenitors which migrate to the brain from blood islands of the yolk sac during early prenatal development (E8) and provide an immune response in neural tissue [10,11]. Unlike macrophages in peripheral tissues microglial cells colonize the brain during embryonic development and continuously renew themselves without replenishment from peripheral blood [11]. Increased levels of proinflammatory cytokines and activation of microglia were found in patients suffering from mental disorders, as well as in animal models of these pathologies (rats, mice) [12]. Microglia can exist in different states. It is supposed that in physiological conditions microglial cells are maintained in a rested (quiescent) state and have ramified morphology with long processes that enable the cells to monitor nervous tissue for pathogen- and damage-associated molecular patterns (PAMPs and DAMPs) [3]. In animal models of psychological and environmental stress microglia undergo morphological and functional changes showing ameboid phenotype with hypertrophic soma and de-ramification [13]. Different types of psychosocial and emotional stressors lead to activation of microglia in the different brain areas of mice and rats and this phenomenon is associated with behavioral abnormalities [3]. These findings are consistent with the hypothesis that exposure to stressors increases the risk of a mental illness through the activation of microglia and other immune responses.

The mechanisms and dynamics of such post-stress neuroimmune changes, their contribution to the pathogenesis of stress-related mental diseases, as well as the relationship of the intensity of inflammation with genetically determined features of the nervous system have not been clarified. Such a characteristic of the nervous system as excitability is of special interest. At the Pavlov Institute of Physiology of Russian Academy of Sciences two strains of rats with contrasting excitability were created as a result of long-term selection [14]. It was shown that rat strains differ in their behavioral responses to stress, and are also characterized by distinct molecular and cytological features of the nervous tissue in control and post-stress states. In HT strain rats (high excitability threshold, low excitable) long-term stressor caused depressive-like behavioral symptoms, increasing of excitability and aggressiveness, while in rats of the LT strain (low excitability threshold, high excitable)—the emergence and preservation of the stereotypical compulsive movements [14]. The genetically determined level of excitability of the nervous system also affects epigenetic milieu that have different dynamics in the brain areas associated with emotions. Therefore, low and high levels of excitability can be considered as risk factors for the development of post-stress pathologies such as PTSD and compulsive disorders [15]. The study of the influence of the genetically determined level of nervous system excitability on the development of immune dysfunctions in response to stressors can help us to understand the nature of individual differences in the pathogenesis of post-stress mental disorders. Thus, the goal of this work is to evaluate the dynamics of post-stress inflammation

in the peripheral blood and hippocampus of rats of two strains which differ in the level of excitability of the nervous system.

## Materials and methods

### Animals and stress protocol

The experiments were performed on 5-month-old adult males of two rat strains differing in excitability of the peripheral and central nervous systems [14,16], which were selected over 80 generations for a high threshold (HT strain) and low threshold (LT strain) of excitability of the tibial nerve (n. tibialis) to electric current.

The strains are included in the biocollection of the I.P. Pavlov Institute of Physiology, RAS (No. GZ 0134-2018-0003), patents for selection invention No. 10769 and 10768 issued by the State Commission of the Russian Federation for Testing and Protection of Selection Inventions, registered in the state register of protected selection inventions on January 15, 2020.

The source material was outbred of Wistar albino rats (breeding nursery Rappolovo, Leningrad region). The selection was carried out according to the value of the threshold of neuromuscular excitability in a test of electric shock irritation of the tibial nerve—*n. tibialis* (rectangular electrical impulses with a duration of 2 ms). The value of the voltage at which the motor reaction appeared was evaluated.

In the first 2 generations full siblings were crossed. intrastrain breeding was carried out in a random order starting from the 3rd generation. Since the 10th generation, breeding has reached a plateau. At the same time, the 4-fold differences between the strains significantly exceeded the intrastrain variability [14].

All animals were maintained in standard environmental conditions (23 ± 2°C; 12 h/12 h dark/light cycle) with water and food *ad libitum* in the animal care facility at Pavlov Institute of physiology of Russian Academy of Sciences. All animal experiments were conducted in accordance with the Council of the European community directives (86/609/EEC) on the use of animals for experimental research. The protocol was approved by the Commission on humane treatment of animals of Pavlov Institute of physiology of Russian Academy of Sciences. Experimental animals were subjected to psychogenic long-term emotional and painful stress according to the K. Hecht protocol [14]: every day for consecutive 15 days animals were subjected to 6 unsupported (10 seconds each) and 6 current-reinforced (2.5 mA, 2 ms) light signals. According to the scheme, the combinations of conditional and unconditional stimuli were not repeated but alternated with a probability of 0.5 which did not allow the animals to develop a conditioned reflex.

To avoid the effect of anesthesia on the profile of proinflammatory signals we used instant decapitation of animals with a guillotine. The manipulation was carried out by an experienced laboratory assistant. Stressed animals and corresponding control groups were sacrificed at different time-points for collecting material (brain) for immunohistochemical staining (Fig 1B): 1) 24 hours after stress exposure; 2) 7 days after stress exposure; 3) 24 days after stress exposure. Blood smears were taken at the same time points as well as before the start of the stress protocol (Fig 1B). In order to minimize the impact of previous behavioral tests on animals OF and EPM were performed on different days (Fig 1A). All control animals were not exposed to stress.

The size of the groups varied. For each test the groups were formed from previously unused animals: behavioral experiments n = 4–12 per group (97 in total), estimation of N:L ratio n = 9–11 per group (43 in total), immunohistochemistry for Iba+ cells in the hippocampus n = 5–6 per group (66 in total). The total number of animals was 206. There was no mortality but in some cases data from animals were excluded due to poor quality of blood smears (1 case) or rats jumping out of experimental chambers during behavioral tests (8 cases).

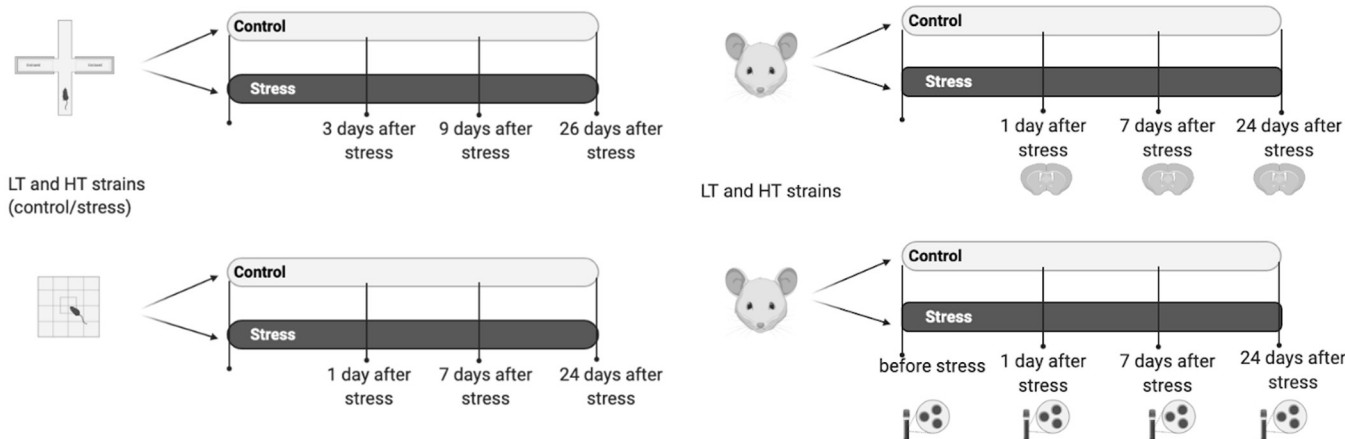

**Fig 1. A timeline of the experiments.** A–behavioral tests, B–blood smears (bottom) and brain immunohistochemistry (top), HT—animals with high excitability threshold, LT–animals with low excitability threshold.

## Behavioral tests

All behavioral tests were performed in a separate facility for behavioral tests in the morning hours. All mazes were thoroughly cleaned with 70% ethanol before the experiments and between animals. The Elevated Plus Maze (EPM) was used to assess anxiety-like behavior. Relevant parameters included the time which animals have spent in the open arms and closed arms of the maze, number of entries into the center, time of grooming(s), supported and unsupported rears. An entry was scored when the midpoint of an animal entered a new compartment; a full entry was defined as the whole animal entering (excluding the tail). Locomotion was measured in an Open Field (OF) arena (1.5 m in diameter and 40 cm high). The latency of the first movement(s), entries to the center, the number of crossed sectors, the duration of immobility(s) and grooming time(s), supported and unsupported rears were calculated. In both tests animals were placed in the test cameras and allowed to move freely for 5 min.

## Evaluation of the dynamics of the neutrophil:lymphocyte ratio

In order to evaluate changes in the neutrophil:lymphocyte ratio four time points were selected: before stress, and 24 hours, 7 days and 24 days after the stress exposure. Blood sampling was made in the morning hours from the tail vein of rats of all control and experimental groups. Neutrophil:lymphocyte ratio was estimated by using blood smear stained with Giemsa and the blood cells phenotype was determined according to the recommendations for hematologic assessment in rats [17].

## Immunohistochemistry

The brains were dissected in ice-cold phosphate-buffered saline (PBS, pH = 7.4) and fixed in 4% paraformaldehyde (PFA) for immunohistochemistry. Brains were further washed several times in PBS, embedded in 5% agarose (Dia-m, LM) and cut on vibrating microtome 7000 (Campden Instruments LTD, UK) in order to obtain 50 μm serial sections. The agarose surrounding each section was removed before immunohistochemistry stainings. Serial sections were placed in 12-well plates in PBS with 0.1% sodium azide. Each strip of sections was then washed in PBS 3 times for 10 min and incubated in a blocking solution containing 3% donkey serum (Abcam, ab7475) and 0.3% Triton X-100 (Sigma-Aldrich, Lot: 2725C289) overnight. Primary antibodies raised in goat against Iba1 were used in the study (Abcam, ab5076). The sections were incubated

with primary antibodies (1:1000) for 48 h, then washed in PBS with 1% donkey serum and subsequently incubated with donkey anti-goat antibodies conjugated to AlexaFluor488 (Abcam, 50129) (1:1000) for 24 h. After the staining all sections were washed 3 times in PBS for 10 min, mounted on glass slides using Fluoroshield mounting medium (Abcam, 104135) and examined under fluorescent microscope Axio imager A2 (Carl Zeiss, Germany) and confocal scanning microscope LSM 780 (Carl Zeiss, Germany) with ZEN software.

Cells were counted at a final magnification of 40×. Three regions of interest (ROI) in the hippocampus were analyzed (CA1, CA3, and DG), and cells were counted on 4 frames in each ROI. The number of microglial cells was calculated on 3 sections from each animal.

### Statistical analysis

Data were analyzed with the BM SPSS Statistics 21 statistical package. The Kolmogorov–Smirnov test was used to analyze the normality of the variables. Kruskal-Wallis test and the two-group Mann-Whitney U test were used for behavioral and immunohistochemistry data analysis and Wilcoxon test and false discovery rate (FDR) to control multiple testing (in case of N:L ratio evaluation). Significance threshold was set at $P \leq 0.05$ for all statistical analyses.

## Results

### Interstrain differences in the threshold voltage levels in LT and HT rat strains

In order to demonstrate the contrast in the level of excitability of the animals from two strains, we evaluated the differences in the voltage threshold at which the motor reaction appeared (Fig 2).

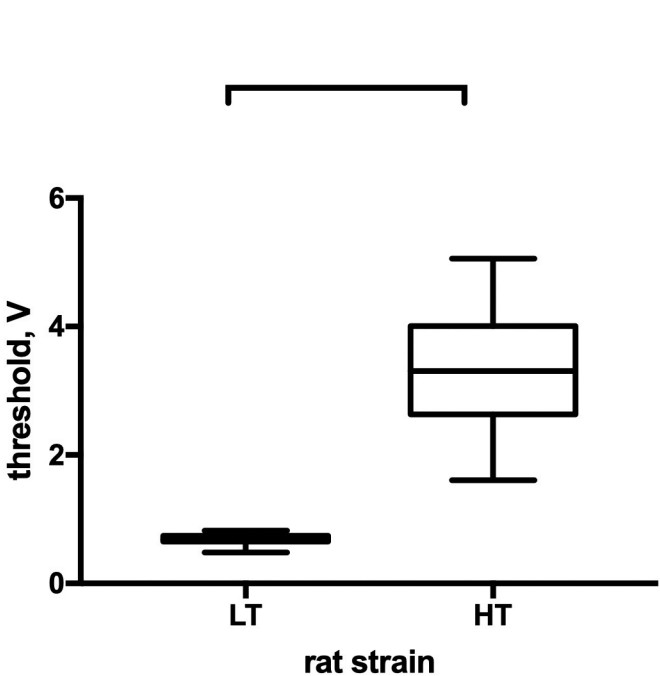

**Fig 2. Interstrain differences in the volage threshold in LT and HT rat strains.** HT—animals with high excitability threshold, LT–animals with low excitability threshold, n = 32–34 per group; the graphs represent the medians, quartile boundaries, and maximum and minimum values of the analyzed data; ** p≤ .01 (Mann-Whitney test).

## Interstrain differences in behavior after the long-term stress exposure

We evaluated the effect of long-term stress on the behavior of LT and HT rats. We did not observe any differences in behavior of HT rats in OF and EPM tests at the studied time points after stressful exposure, while LT rats showed significant behavioral response to stress (Figs 3 and 4).

## Effect of stress on neutrophil:lymphocyte ratio

There were no interstrain differences in N:L ratio (Fig 5A). In response to stress however, a significant change in N:L ratio compared with the control animals was observed in stressed HT rats 24 hours after stress exposure (Fig 5B). No significant differences were found between the LT control and stressed groups (Fig 5C).

In the group of stressed LT rats we observed a significant increase in N:L ratio on 24 h and 7 days after stress exposure (Figs 5E and 6) but there were no significant differences in the LT-control (not present). In the HT group N:L ratio was reduced 24 hours after stress exposure, and then restored to its initial values by day 7 (Fig 5D).

## Effect of stress on microglial activation in the hippocampus

We revealed significant differences in the number of microglial cells in some areas of the hippocampus in HT and LT rats. Interstrain differences were found in the CA1 (Fig 7A) and CA3

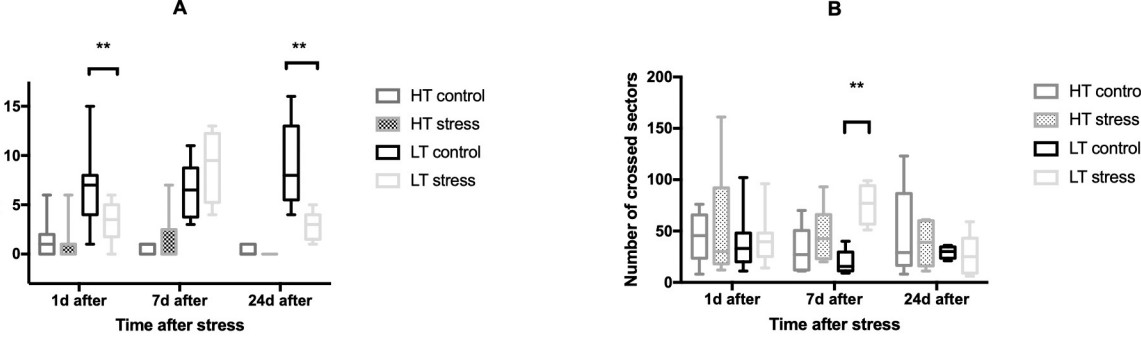

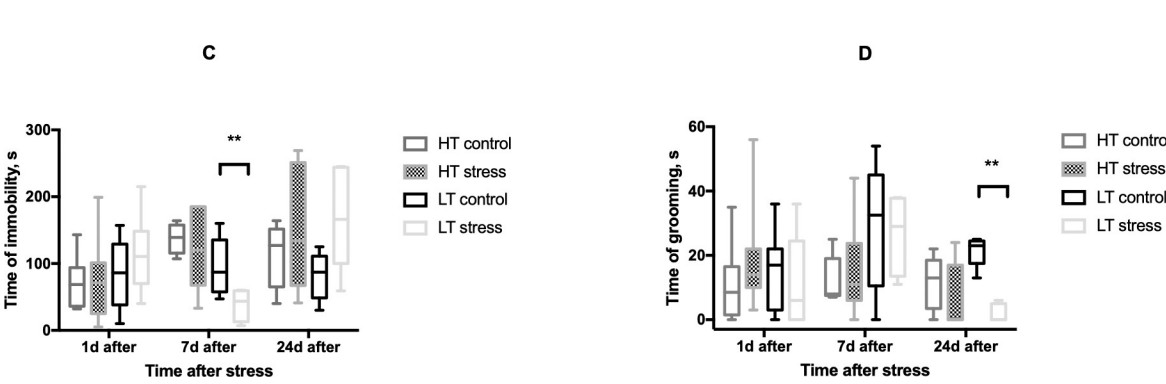

**Fig 3. Effect of long-term stress on open field activity of HT and LT rats.** The number of unsupported rears (A), number of crossed sectors (B), time of immobility (C), time of grooming (D); HT control—animals with high excitability threshold, control group, HT stress—animals with high excitability threshold, stressed group, LT control–animals with low excitability threshold, control group, LT stress–animals with low excitability threshold, stressed animals, n = 4–12 per group; the graphs represent the medians, quartile boundaries, and maximum and minimum values of the analyzed data; ** p≤ .01 (Mann-Whitney test).

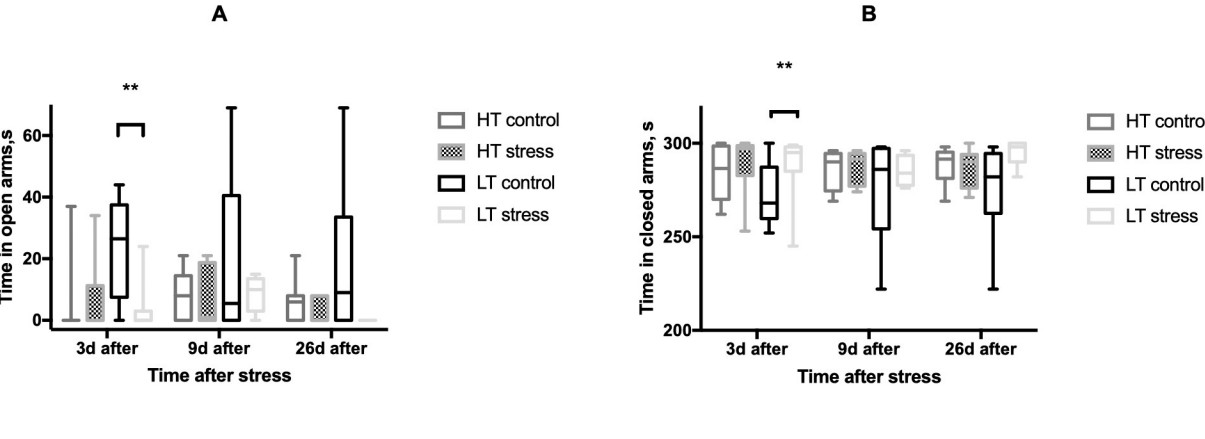

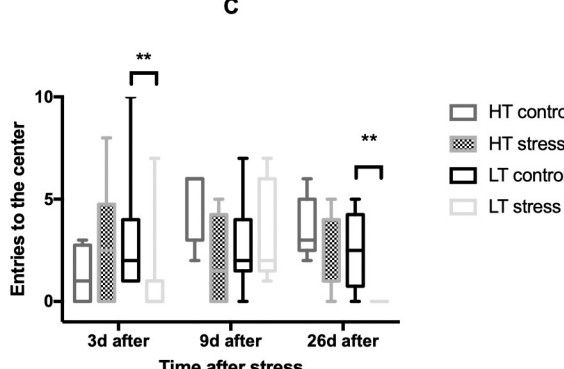

**Fig 4. Effect of long-term stress on elevated plus maze activity of HT and LT rats.** Time spent in open arms (A), time spent in closed arms (B), number of visits to the center of the field (C); HT control—animals with high excitability threshold, control group, HT stress—animals with high excitability threshold, stressed group, LT control–animals with low excitability threshold, control group, LT stress–animals with low excitability threshold, stressed animals, n = 5–12 per group; the graphs represent the medians, quartile boundaries, and maximum and minimum values of the analyzed data; ** p≤ .01 (Mann-Whitney test).

(Fig 7B) areas. LT rats showed significantly lower numbers of Iba1+ cells compared to the HT. Fig 7 shows the number of microglial cells in the hippocampus as visualized by immunohistochemistry.

In HT stressed rats we found a significant increase in microglial cells number only in the CA1 region 7 days after stress (compared to the HT control) (Fig 8A).

In LT rats the number of microglial cells in all studied areas of the hippocampus was significantly increased 7 days after long-term stress (Figs 8 and 9). 24 days after stress exposure, the number of microglial cells did not differ from the control (Fig 8).

## Discussion

Altered excitability of the nervous system is linked to certain disorders, such as anxiety [18] and depression [19], however the pathogenesis of these conditions is complex and not well understood. We used two rat strains which differ in the level of excitability of the nervous system in order to see whether it affects their responses to chronic stress, and we found out that highly excitable LT rats are in fact more vulnerable to stress than less excitable HT animals.

The OF is usually used in order to assess general locomotor activity of an animal while in the EPM one can estimate anxiety-like behavior [20]. According to [21], in EPM and OF high

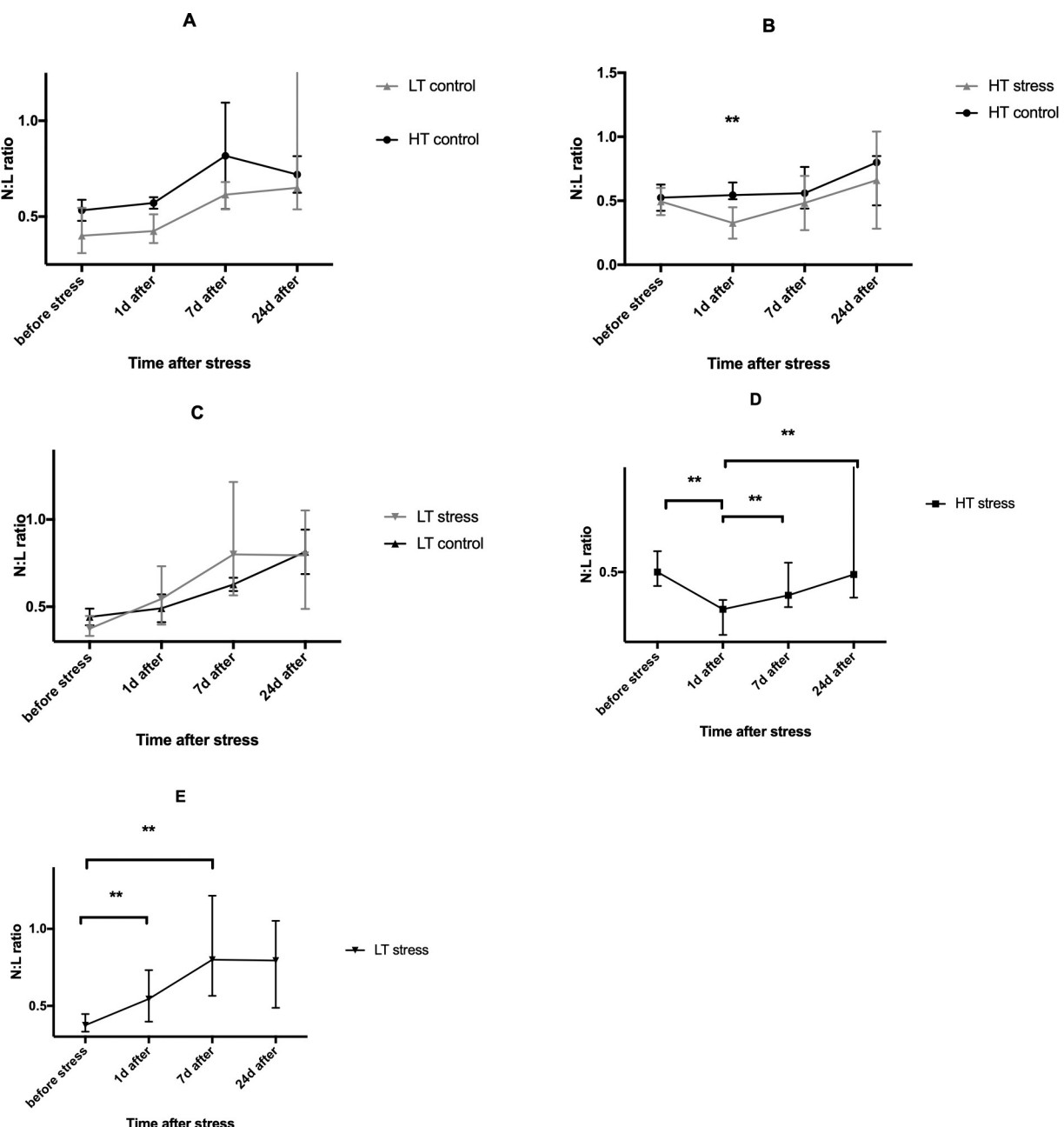

**Fig 5. Effect of long-term stress on neutrophil:lymphocyte ratio (N:L ratio) in LT and HT rats.** Intertrain comparison of intact rats (A), effect of stress on N:L ratio in HT strain (B), effect of stress on N:L ratio in LT strain (C), dynamics of stress related changes in N:L ratio in HT (D) and in LT rats (E). LT control–low excitability threshold, control group, LT stress–low excitability threshold, stressed animals; HT control–high excitability threshold, control group, HT stress–high excitability threshold, stressed animals; n = 9–11 per group; bars represent medians and interquartile range; B—** p≤ .01 (Mann-Whitney test), D-F **—p≤ .01 (Wilcoxon test with FDR correction).

excitable LT rats show a lower level of anxiety and a higher level of exploratory behavior than HT rats.

In response to stress animals show significant differences in behavior. We did not observe any stress-related behavioral effects in HT animals while in LT rats stress-induced behavioral changes had complex dynamics. One and three days after stress exposure exploratory behavior

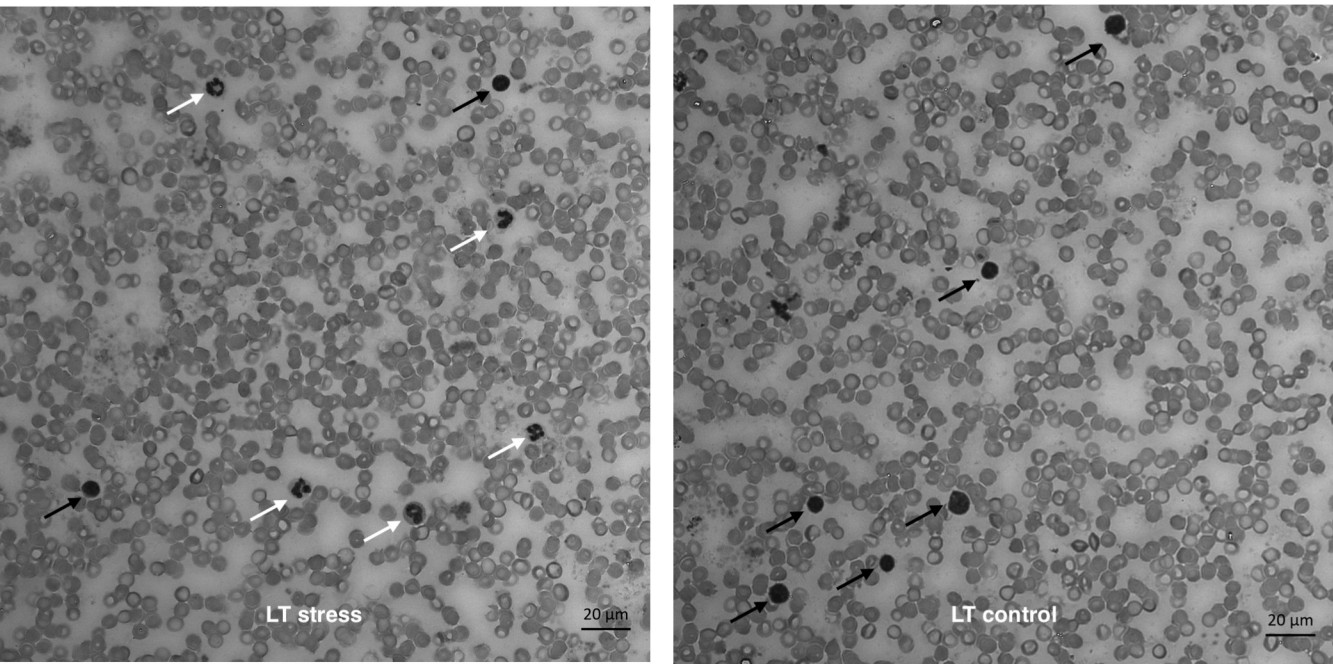

**Fig 6.** Neutrophils (white arrows) and lymphocytes (black arrows) in LT group before stress (right) and LT group 7 days after stress (left).

was significantly reduced, particularly we observed a decrease in the number of unsupported rears in OF, the time spent in the open arms of the EPM, and the number of entries to the center of the maze. This can be interpreted as an increase in anxiety-like behavior which was not the case for HT rats. In the OF LT rats showed increased nonspecific locomotor activity 7 days after stress exposure, but surprisingly anxiety-like symptoms in the EPM test disappeared. 24–26 days after stress exposure exploratory behavior of LT rats was again significantly reduced (reduced grooming time, the number of unsupported rears in the OF and reduced entries to the center in the EPM). Thus, despite that LT rats normally show less anxiety in OF and EPM than HT animals they are more vulnerable to chronic stress at the studied time points after stressful exposure. It is known that activation of medial prefrontal cortex (mPFC) stimulates the amygdala and it leads to anxiety-like behavior in rats. Bi et al. [22] demonstrated that electrolytic lesions of the rodent mPFC decrease anxiety-like behavior in the EPM and increase locomotion in OF. There is evidence that LT rats have more excitable amygdala, since they show a greater frequency of amygdala impulses in response to electrical stimulation of the infralimbic PFC [16]. Perhaps disturbances in the functioning of the PFC or its connections with the amygdala are the basis for anxiety-like behavior of LT rats in response to stress. The disappearance of anxiety behavior and an increase in locomotion 7 days after stress exposure may be associated with the involvement of compensatory mechanisms in the central nervous system. However the cellular and molecular identity of these mechanisms is not clear and apparently these mechanisms are not enough since we observed anxiety-like symptoms in LT animals later again.

Animals react to stress by activating hypothalamic pituitary-adrenocortical axis (HPA) and releasing glucocorticosteroids from the adrenal cortex. Among many functions glucocorticosteroids show heterogeneous effects on immune cells: increased survival and accumulation of neutrophils at inflammatory sites while induction of apoptosis in T- and B-lymphocytes [23,24]. It has been shown that glucocorticosteroids (GCs) prevent apoptosis of neutrophils.

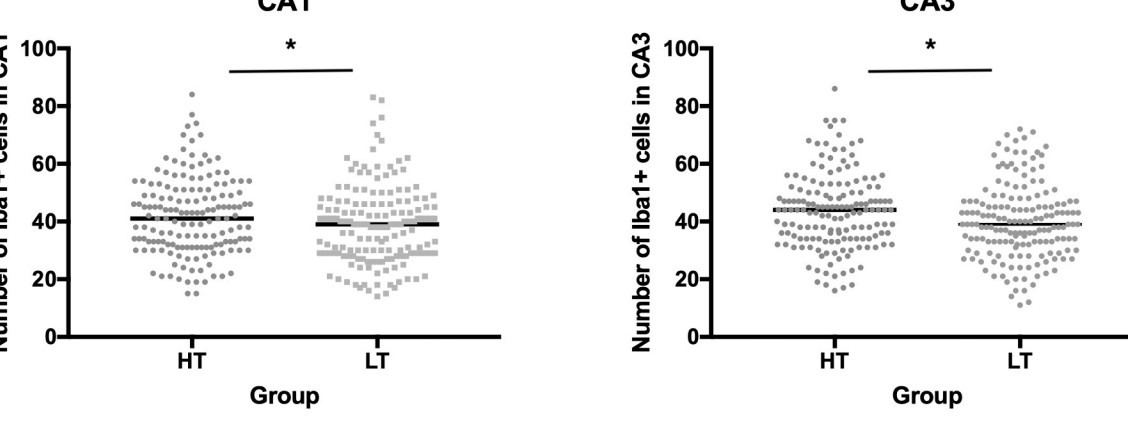

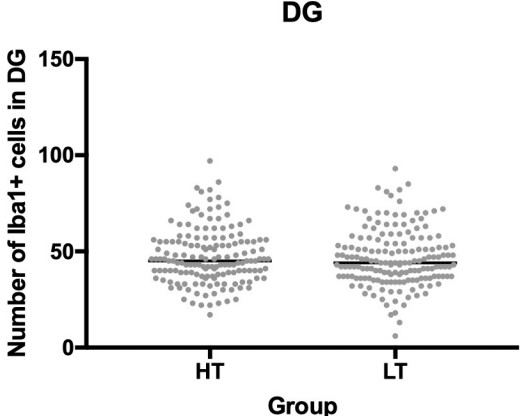

**Fig 7. Interstrain differences in the number of Iba1+ cells in different hippocampal areas.** The number of Iba1+ cells in the CA1, CA3 and DG areas of the hippocampus; n = 5–12 per group; bars represent medians; *p≤0,05 Mann-Whitney U test.

Stress or administration of pharmacological doses of GCs cause neutrophilia, which is dependent on both the increased migration of neutrophils from the bone marrow to blood flow and their increased survival [25]. Thus, stress hormones shift the N:L ratio towards an increase in the number of neutrophils. It is known that the N:L ratio in the peripheral blood correlates with other markers of inflammation (cytokines, chemokines), and therefore the leukocyte shift index can be used as a convenient indicator of both peripheral inflammation and chronic stress [26]. Recent studies show that it can be a marker of the inflammatory process that accompanies some mental disorders (depression, bipolar disorder) [26,27]. There is evidence that in the case of chronic stress, a shift in the leukocyte formula is a more reliable marker of stress than the level of glucocorticoids in the blood. It has been shown that in mice, rats and birds long-term stress can lead to "adaptation" of the HPA and decrease in the level of glucocorticoids in the blood, while an increase in N:L ratio is still observed even a week after the end of a stressor [28]. We did not observe any interstrain differences in N:L ratio in LT and HT rats. Thus, intact animals of both strains do not differ in the initial ratios of neutrophils and lymphocytes in the peripheral blood. In response to stress a change in N:L ratio compared with the control was observed only in the group of stressed HT animals 24 hours after stress exposure. In the group of stressed LT rats there was a significant increase in N:L ratio 1 and 7

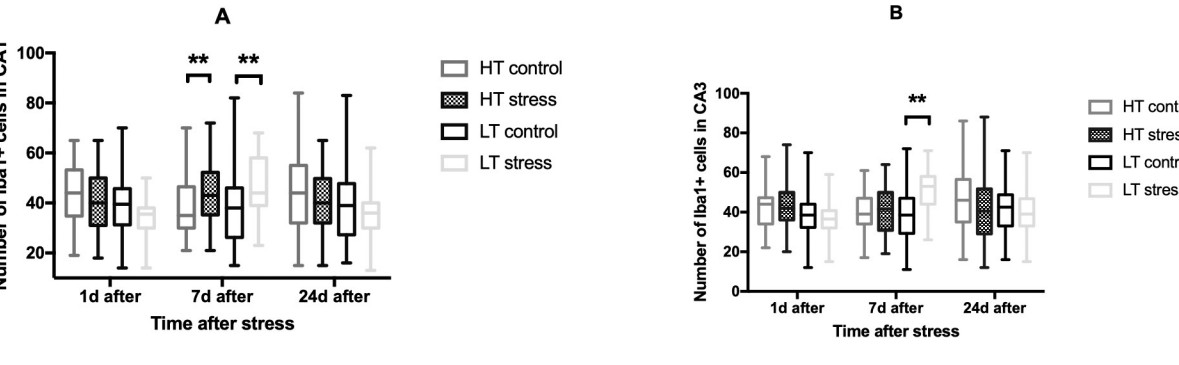

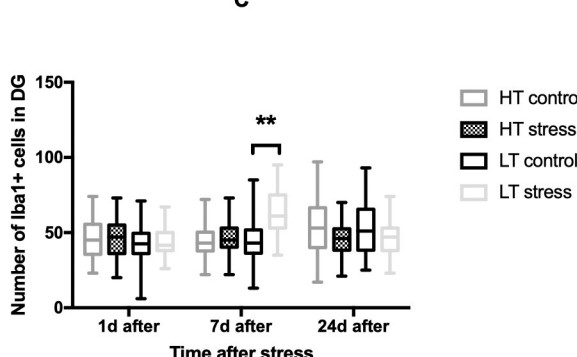

**Fig 8. Effect of long-term stress on the number of microglial cells in the hippocampus in HT and LT rats.** (A) The number of Iba1+ cells in the CA1 area of the hippocampus, (B)–in CA3 area and (C)–in the dentate gyrus (DG); n = 5–12 per group; bars represent medians and interquartile range; **p≤0,01 Mann-Whitney U test.

days after stress exposure, and in the HT rats stress reduced N:L ratio 24 hours after stress and restored the ratio to its initial values on day 7.

Animals of LT strain are characterized by higher excitability not only of the tibial nerve (the main feature for selection), but also of the reticular formation of the midbrain, as well as the amygdala which is associated with emotional regulation and stress response [14,16]. It is known that subcutaneous administration of corticosterone in mice for 14 days leads to hyper-activity of the amygdala, as well as to disturbances of amygdala-mediated memory of a trau-matic event [29]. Previously, it was shown that strains of highly excitable rats with a low threshold of sensitivity to electric current are characterized by increased stress reactivity of the HPA, accelerated development of the hormonal response, and reduced sensitivity of the HPA to feedback signals [30]. It makes animals of this strain more susceptible to immune dysfunc-tions in response to chronic stress and may lead to excessive activation of the amygdala.

We found that intact animals of the highly excitable LT strain had significantly less Iba + microglial cells in the CA1 and CA3 regions of the hippocampus than the less excitable HT strain. It can be assumed that in the absence of a stressor, the immune defence of the hippo-campal tissue in highly excitable animals is weakened. Microglial cells are known to be at rest-ing state under physiological conditions and scan the environment for proinflammatory signals [3]. These cells also help to regulate brain function by removing dying neurons, prun-ing non-functional synapses, and producing signals that support neuronal survival. There is

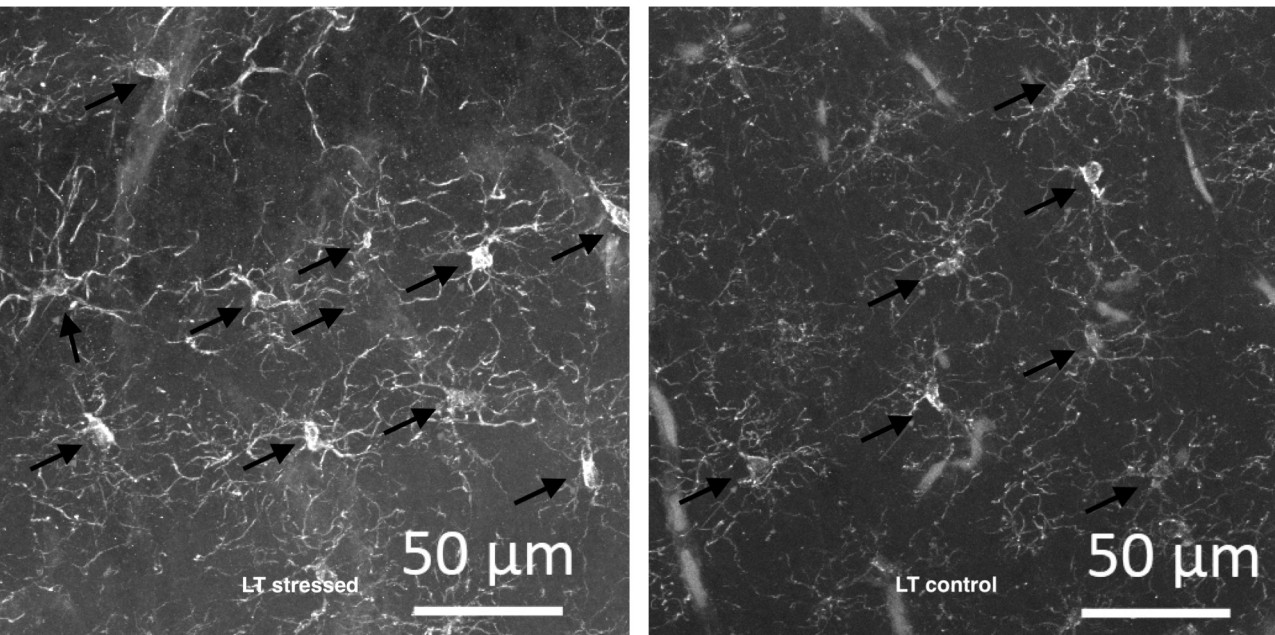

**Fig 9.** Iba1+ cells in the DG of LT stressed group (left) and LT control group (right).

evidence that microglial cells are also critical modulators of neuronal activity and corresponding behavioral responses in mice. Microglia respond to neuronal activation by suppressing neuronal activity [31]. So, it is possible that a reduced number of microglia in the hippocampus of highly excitable animals is one of the reasons for hyperexcitability of their nervous systems (for example, the amygdala) and specific behavioral activity in tests of OF and EPM.

No changes in microglial numbers were identified in stressed HT rats compared to control with the exception of a significant increase in cell number in the CA1 region 7 days after stress (compared to the HT control). But in a strain of highly excitable LT rats, we observed a significant increase in the number of Iba+ cells 7 days after stress exposure in all studied areas of the hippocampus. In order to determine the phenotype of these cells (whether they belong to activated microglia or microglial cells "at rest"), it is necessary to study other markers of neuroinflammation, such as the level of pro-inflammatory interleukins, as well as to evaluate morphology of cells. But from our data, we can conclude that at least the response of microglial cells to stress in highly excitable animals is more generalized and manifests in all studied areas of the hippocampus. Previous reports showed that activated microglia stimulate the presynaptic release of excitatory transmitters. It was shown that activated microglia also alter non-synaptic membrane excitability in neurons and synaptic transmission via tumor necrosis factor in the neocortex, cerebellum and hippocampus [32]. Perhaps the behavioral abnormalities that we observed in response to stress in highly excitable animals are partly related to generalized activation of microglia in all areas of the hippocampus. For a detailed understanding of the effect of CNS excitability on post-stress neuroinflammation it is necessary to study microglial alterations in other brain regions that are relevant to stress such as amygdala and PFC.

## Conclusion

Thus, hereditary high excitability of the nervous system is a possible risk factor for the development of post-stress pathologies since:

- Highly excitable animals show a decrease in exploratory behavior in OF and EPM tests (in 1–3 and 24–26 days after stress) and an increase (for 7 days) in locomotion in OF in response to long-term stress.

- In response to stress N:L ratio in the highly-excitable rats increases 24 hours after stress exposure and this increase persists for up to 7 days after stress.

- High excitability is associated with a significant increase in the number of microglial cells in all studied areas of the hippocampus 7 days after stress.

## Supporting information

**S1 Table. The direction of poststress changes in the studied parameters in rats of two strains.**
(DOCX)

**S1 File. Volage threshold data in LT and HT rat strains.**
(PZFX)

**S2 File. Activity in Open Field data.**
(PZFX)

**S3 File. Activity in Elevated Plus Maze data.**
(PZFX)

**S4 File. Data of N:L ratio.**
(PZFX)

**S5 File. Number of microglial cells in hippocampus (data).**
(PZFX)

## Author Contributions

**Conceptualization:** O. P. Tuchina, N. A. Dyuzhikova.

**Formal analysis:** I. G. Shalaginova, M. V. Sidorova, A. S. Levina, A. I. Vaido.

**Funding acquisition:** I. G. Shalaginova, N. A. Dyuzhikova.

**Investigation:** I. G. Shalaginova, O. P. Tuchina, M. V. Sidorova, A. S. Levina, D. A-A. Khlebaeva, A. I. Vaido.

**Methodology:** O. P. Tuchina, M. V. Sidorova, A. S. Levina, D. A-A. Khlebaeva, A. I. Vaido.

**Project administration:** I. G. Shalaginova.

**Supervision:** O. P. Tuchina, N. A. Dyuzhikova.

**Visualization:** I. G. Shalaginova.

**Writing – original draft:** I. G. Shalaginova.

**Writing – review & editing:** O. P. Tuchina, A. S. Levina, N. A. Dyuzhikova.

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
