## [Decision Letter · Decision Letter 0]

17 May 2021

PONE-D-21-11095

Effects of psychogenic stress on some peripheral and central inflammatory markers in rats with the different level of excitability of the nervous system

PLOS ONE

Dear Dr. Shalaginova,

Thank you for submitting your manuscript to PLOS ONE. After careful consideration, we feel that it has merit but does not fully meet PLOS ONE’s publication criteria as it currently stands. Therefore, we invite you to submit a revised version of the manuscript that addresses the points raised during the review process.

We look forward to receiving your revised manuscript.

Kind regards,

Alexandra Kavushansky, PhD

Academic Editor

PLOS ONE

Journal Requirements:

2. To comply with PLOS ONE submissions requirements, please provide methods of sacrifice in the Methods section of your manuscript.

Reviewers' comments:

Reviewer's Responses to Questions

**Comments to the Author**

1. Is the manuscript technically sound, and do the data support the conclusions?

Reviewer #1: Partly

Reviewer #2: Yes

2. Has the statistical analysis been performed appropriately and rigorously? 

Reviewer #1: I Don't Know

Reviewer #2: Yes

3. Have the authors made all data underlying the findings in their manuscript fully available?

Reviewer #1: No

Reviewer #2: Yes

4. Is the manuscript presented in an intelligible fashion and written in standard English?

Reviewer #1: Yes

Reviewer #2: Yes

5. Review Comments to the Author

Reviewer #1: The manuscript for review by Shalaginova et al. describes the affect of post-stress in rats that were previously selectively bred to retain the traits of low and high excitability thresholds. The authors subjected these 2 strains of rats to a stress protocol and measured the effect on behavior, Iba+ cells in the hippocampus, and neutrophil:lymphocyte ratios in peripheral blood smears. Their results led them to the interpretation that high excitability of the nervous system is a possible risk factor for the development of post-stress pathologies.

Comments:

Page 2, top paragraph. The authors mis-reference the pro-inflammatory cytokine in patients’ studies. The cited work (reference 5) discusses insights from animal models only. It has nothing to do with human subjects. Please include the primary references for this section. Moreover, the authors need to better reference the entire manuscript as a whole.

Page 2, 3rd paragraph. The authors make the incorrect statement that “Microglial cells originate from macrophages.” This needs to be clarified. Microglia are derived from primitive myeloid progenitors that arise before embryonic day 8. Furthermore, the authors need to better references this section. (Refer to the following manuscripts for clarification. Ginhoux F, Greter M, Leboeuf M, Nandi S, See P, Gokhan S, et al. Fate Mapping Analysis Reveals That Adult Microglia Derive from Primitive Macrophages. Science. 2010;330(6005):841-845. AND Perdiguero EG, Klapproth K, Schulz C, Busch K, Azzoni E, Crozet L, et al. Tissue-resident macrophages originate from yolk-sac-derived erythro-myeloid progenitors. Nature. 2015;518(7540):547-551.)

It would be beneficial for all of the complete statistical analyses to be submitted as an excel spreadsheet as supplemental data so that the reader can verify the dataset.

In the Conclusion, page 13. The first point that “Highly excitable animals show a persistent (for 24 days) decrease in locomotion” is an overstatement based on the presented data. The results are not that clean and concise, and without access to transparent statistics (the raw data) it is difficult to fully assess this statement. This should be rectified.

Minor:

Line numbers and page numbers should be included for ease of reviewing.

Page 5, 4th line from top. Need to add “3)” before 24 days.

Check for proper punctuation in the entire manuscript.

Reviewer #2: It is a very nice manuscript and important to the field. Effects of psychogenic stress on some peripheral and central inflammatory markers in rats with the different level of excitability of the nervous system.

1. Please identify the member family of TNF in this statement, or the author is referring to the TNF as a superfamily "tumor necrosis factor (TNF) are significantly increased in the blood of patients with posttraumatic stress disorder (PTSD)."

2. Animals and stress protocol. Please add the animal's age and the Wistar strain in the first line of the paragraph to facilitate the reader. The number of animals per group, the total number of animals, and if they had mortality.

3. I suggest to the authors to add a timeline.

4. I suggest to the author add the details of the levels of CNS excitability of the animals. The value of that intensity above which the neuron spikes (top) and the voltage threshold.

5. Why did the LT rat's anxiety-like symptoms in the EPM test disappeared?

6. Animals react to stress by activating the hypothalamic-pituitary-adrenocortical axis (HPA) and releasing glucocorticosteroids from the adrenal cortex. What is the mechanism that stress increases the number of neutrophils?

7. Did the author evaluate cytokines/chemokines and other inflammatory mediators in the animals.

8. Why did the microglial cells increased the number by are in the resting morphology?

9. Why did the microglial cells increased in number but remained in the resting shape?

10. I suggest a flow chart to explain the conclusion.

6. PLOS authors have the option to publish the peer review history of their article (what does this mean?). If published, this will include your full peer review and any attached files.

Reviewer #1: No

Reviewer #2: No

---

## [Author Response · Author response to Decision Letter 0]

24 Jun 2021

Reviewer 1

1. Page 2, top paragraph. The authors mis-reference the pro-inflammatory cytokine in patients’ studies. The cited work (reference 5) discusses insights from animal models only. It has nothing to do with human subjects. Please include the primary references for this section. Moreover, the authors need to better reference the entire manuscript as a whole.

Thank you for the clarification, the cited article is really not the original source, but contains a link to these works. It is more correct to quote a review that contains summary data about Immunological Factors Associated with PTSD in human.

2. Page 2, 3rd paragraph. The authors make the incorrect statement that “Microglial cells originate from macrophages.” This needs to be clarified. Microglia are derived from primitive myeloid progenitors that arise before embryonic day 8. Furthermore, the authors need to better references this section

Thank you for your comments and recommended papers, there is definitely a mistake in this sentence. We have made appropriate corrections.

3. It would be beneficial for all of the complete statistical analyses to be submitted as an excel spreadsheet as supplemental data so that the reader can verify the dataset.

Since the data analysis was performed in spss and prism, we attach the data in the “pzfx” (prism) format as supplement files, where you can find the raw data, medians, and quartile range.

4. In the Conclusion, page 13. The first point that “Highly excitable animals show a persistent (for 24 days) decrease in locomotion” is an overstatement based on the presented data. The results are not that clean and concise, and without access to transparent statistics (the raw data) it is difficult to fully assess this statement. This should be rectified.

The erroneous statement has been rectified

Minor:

Line numbers and page numbers should be included for ease of reviewing.

Done

Page 5, 4th line from top. Need to add “3)” before 24 days.

Done

Check for proper punctuation in the entire manuscript. Done

Reviewer 2

1. Please identify the member family of TNF in this statement, or the author is referring to the TNF as a superfamily "tumor necrosis factor (TNF) are significantly increased in the blood of patients with posttraumatic stress disorder (PTSD)."

Done

2. Animals and stress protocol. Please add the animal's age and the Wistar strain in the first line of the paragraph to facilitate the reader. The number of animals per group, the total number of animals, and if they had mortality.

Done

3. I suggest to the authors to add a timeline.

Done (Fig 1)

4. I suggest to the author add the details of the levels of CNS excitability of the animals. The value of that intensity above which the neuron spikes (top) and the voltage threshold.

We have added the necessary explanations to the methods section. Also in the Results section, we added a picture showing significant differences in the threshold volage levels in these two rat strains.

The study of evoked potentials in brain structures after electrical stimulation of the tibial nerve is a special task. But we have recently demonstrated contrasting differences between the strains in terms of the evoked neuronal activity of the amygdaloid complex (Prefrontal cortex electrical stimulation). Link to the article by Sivachenko, 2020 is given.

5. Why did the LT rat's anxiety-like symptoms in the EPM test disappeared?

From the data we obtained, it is impossible to conclude about the reason for such dynamics in behavior, but we made an assumption about the involvement of compensatory mechanisms (added to the discussion).

6. Animals react to stress by activating the hypothalamic-pituitary-adrenocortical axis (HPA) and releasing glucocorticosteroids from the adrenal cortex. What is the mechanism that stress increases the number of neutrophils?

Endogenous GCs are one of the factors that promote the maturation of neutrophils in the bone marrow and favor the mobilization of neutrophils from the bone marrow into circulation. Another mechanism is neutrophil resistance to GC-induced apoptosis, which prolongs their life span. We have added an explanation to the discussion text.

7. Did the author evaluate cytokines/chemokines and other inflammatory mediators in the animals.

Of course, it is necessary to evaluate other markers of neuroinflammation. We plan to evaluate the level of proinflammatory cytokines in the brain and blood of these animals in our next studies.

8. Why did the microglial cells increased the number by are in the resting morphology?

Based on our data, we cannot draw conclusions about the state of the microglia (activated\\resting).

9. Why did the microglial cells increased in number but remained in the resting shape?

From our data, we cannot conclude which phenotype (activated or resting) has microglia in the hippocampus. In order to answer this question, it is necessary to study other markers of neuroinflammation (cytokines) and the morphology of microglial cells. We noted this in the discussion.

10. I suggest a flow chart to explain the conclusion.

The table “The direction of poststress changes in the studied parameters in rats of two strains” was added in Supplement (S6).

---

## [Decision Letter · Decision Letter 1]

15 Jul 2021

Effects of psychogenic stress on some peripheral and central inflammatory markers in rats with the different level of excitability of the nervous system

PONE-D-21-11095R1

Dear Dr. Shalaginova,

We’re pleased to inform you that your manuscript has been judged scientifically suitable for publication and will be formally accepted for publication once it meets all outstanding technical requirements.

Kind regards,

Alexandra Kavushansky, PhD

Academic Editor

PLOS ONE

Additional Editor Comments (optional):

Reviewers' comments:

Reviewer's Responses to Questions

**Comments to the Author**

1. If the authors have adequately addressed your comments raised in a previous round of review and you feel that this manuscript is now acceptable for publication, you may indicate that here to bypass the “Comments to the Author” section, enter your conflict of interest statement in the “Confidential to Editor” section, and submit your "Accept" recommendation.

Reviewer #1: All comments have been addressed

Reviewer #2: All comments have been addressed

2. Is the manuscript technically sound, and do the data support the conclusions?

Reviewer #1: Yes

Reviewer #2: Yes

3. Has the statistical analysis been performed appropriately and rigorously? 

Reviewer #1: Yes

Reviewer #2: Yes

4. Have the authors made all data underlying the findings in their manuscript fully available?

Reviewer #1: Yes

Reviewer #2: Yes

5. Is the manuscript presented in an intelligible fashion and written in standard English?

Reviewer #1: Yes

Reviewer #2: Yes

6. Review Comments to the Author

Reviewer #1: (No Response)

Reviewer #2: The authors improved the manuscript, correct the text, adding more information to facilitate the readers.

7. PLOS authors have the option to publish the peer review history of their article (what does this mean?). If published, this will include your full peer review and any attached files.

Reviewer #1: No

Reviewer #2: No

---

## [Editor Report · Acceptance letter]

19 Jul 2021

PONE-D-21-11095R1 

Effects of psychogenic stress on some peripheral and central inflammatory markers in rats with the different level of excitability of the nervous system 

Dear Dr. Shalaginova:

I'm pleased to inform you that your manuscript has been deemed suitable for publication in PLOS ONE. Congratulations! Your manuscript is now with our production department. 

Kind regards, 

on behalf of

Dr. Alexandra Kavushansky 

Academic Editor

PLOS ONE